# DIFFUSION IMPLICIT POLICY FOR UNPAIRED SCENE-AWARE MOTION SYNTHESIS

## ABSTRACT

Human motion generation is a long-standing problem, and scene-aware motion synthesis has been widely researched recently due to its numerous applications. Prevailing methods rely heavily on paired motion-scene data whose quantity is limited. Meanwhile, it is difficult to generalize to diverse scenes when trained only on a few specific ones. Thus, we propose a unified framework, termed Diffusion Implicit Policy (DIP), for scene-aware motion synthesis, where paired motion-scene data are no longer necessary. In this framework, we disentangle human-scene interaction from motion synthesis during training and then introduce an interaction-based implicit policy into motion diffusion during inference. Synthesized motion can be derived through iterative diffusion denoising and implicit policy optimization, thus motion naturalness and interaction plausibility can be maintained simultaneously. The proposed implicit policy optimizes the intermediate noised motion in a GAN Inversion manner to maintain motion continuity and control keyframe poses though the ControlNet branch and motion inpainting. For long-term motion synthesis, we introduce motion blending for stable transitions between multiple sub-tasks, where motions are fused in rotation power space and translation linear space. The proposed method is evaluated on synthesized scenes with ShapeNet furniture, and real scenes from PROX and Replica. Results show that our framework presents better motion naturalness and interaction plausibility than cutting-edge methods. This also indicates the feasibility of utilizing the DIP for motion synthesis in more general tasks and versatile scenes.

## 1 INTRODUCTION

Synthesizing human motion in real 3D scenes has attracted significant attention in recent years Cao et al. (2020); Wang et al. (2021; 2022a); Zhao et al. (2023), due to its wide applications in scene simulation, digital human animation, and virtual/augmented reality.

Thanks to learning-based 3D perception Qi et al. (2017); Zhao et al. (2021), pioneers Cao et al. (2020); Wang et al. (2021); Zhao et al. (2023) have attempted to synthesize motion in scenes with feasible human-scene interaction. However, in almost all previous works, paired motion-scene data are required to learn scene-aware motion policies. The majority of prevailing methods Starke et al. (2019); Zhao et al. (2023) learn **Explicit Policies** to directly predict the desired motion based on current states and goals (Fig. 1 (a)). Some of them Wang et al. (2021; 2022a) utilized a second-stage **Implicit Policy** optimization but sacrifice the motion naturalness for interaction plausibility (Fig. 1 (b)). Recent works Huang et al. (2023); Wang et al. (2024) utilized conditional **Diffusion Policies** to achieve better performance (Fig. 1 (c)), where massive paired motion-scene data is also necessary.

In fact, captured human motion data Mahmood et al. (2019); Lin et al. (2023) is far more abundant than paired motion-scene data Hassan et al. (2019); Wang et al. (2022b). Motion synthesis that relies heavily on paired data will inevitably suffer from limited diversity. Meanwhile, the generalization ability is hard to guarantee when trained on limited scenes and applied to various other scenes.

Based on this observation, we propose a unified framework, termed **Diffusion Implicit Policy (DIP)** (Fig. 1 (d)), which disentangles human-scene interaction from motion synthesis during training and then integrate motion denoising with implicit policy optimization during inference. In this way, paired motion-scene data is no longer necessary for training, and motion naturalness and interaction plausiblity can be ensured simultaneously for scene-aware motion synthesis.

In the DIP, a motion diffusion model is employed to make the synthesized motion more and more natural throughout the entire denoising process(Fig. 2 (b)). We equipped the diffusion model with a ControlNet Zhang et al. (2023a) branch to provide keyframe joint hints for historical motion and future goals. Following previous works Tevet et al. (2023); Xie et al. (2024), the diffusion model is designed to predict the original motion at each denoising step and then sample the denoised motion from a normal distribution accordingly. Thus, we can well utilize the stochastic process to pursue plausible human-scene interactions. Specifically, interaction-based reward functions are designed to assess the consistency between motions and scenes. These reward functions are used as implicit policy to optimize the sampling distribution, ensuring that the sampled denoised motion corresponds better to the 3D scenes at each denoising step. As the denoising process can also be viewed as optimization for motion naturalness, the entire scene-aware motion synthesis can be framed as an optimization problem to simultaneously pursue both motion naturalness and interaction plausibility.

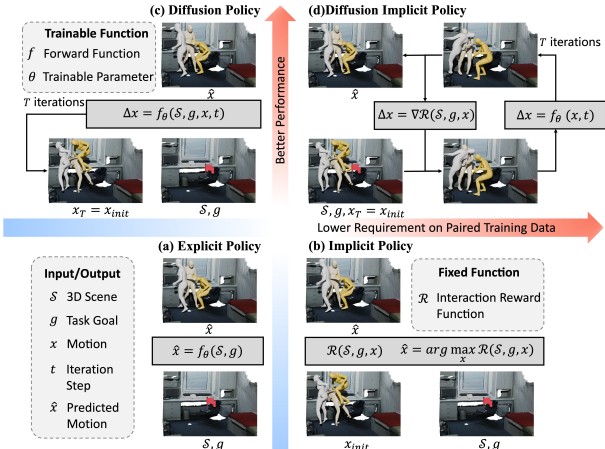

Figure 1: Policy learning frameworks. (a) Explicit policy is trained with paired motion-scene data given task to predict the final motion. (b) Implicit policy optimizes the motion from initialization accordingly. (c) Diffusion policy gradually denoise the motion based on current scene, task, and noised motion. (d) Our diffusion implicit policy iteratively denoises and optimizes the motion to ensure motion naturalness, diversity, interaction plausibility simultaneously without need for any paired data.

To synthesize reasonable motion in 3D scenes, we first train a motion diffusion model conditioned on actions and keyframe joints, which can be derived from motion itself. Furthermore, we design various reward functions to score motion naturalness and interaction plausibility. These rewards will optimize the sample distribution during motion denoising. We choose to adjust the centroid of the distribution in a GAN inversion manner, applying these reward functions to the outputs of the diffusion model at the centroid rather than directly to the centroid itself. In this way, the proposed method can identify a better intermediate noised motion with higher motion naturalness and interaction plausibility in the final synthesized motions.

In addition, for long-term motion synthesis involving multiple tasks, we need to take historical motion as constrain when synthesizing future motion. To maintain continuity between historical and future motions, we employ a time-variant motion blending, where we interpolate the rotation matrix in the power space and the translation in standard linear space. Thus far, the proposed framework can synthesize long-term motion in general scenes without any training on paired motion-scene data.

For performance evaluation, we use scenes cluttered with furniture from ShapeNet Chang et al. (2015) to assess the ability on human-object interaction. We also take PROX Hassan et al. (2019) and Replica Straub et al. (2019) to demonstrate the generalization ability in scene-aware motion synthesis. We compared the proposed method with prevailing works based on physical and perceptual scores. Comprehensive experiments support our claims and indicate that the synthesized motion produced by the proposed method demonstrates better performance.

Our main contributions can be summarized as follows: (1) We propose a brand-new framework, termed Diffusion Implicit Policy, for scene-aware motion synthesis. In this framework, we disentangle human-scene interaction from motion synthesis during training and transform scene-aware motion synthesis into a joint optimization problem, where motion naturalness and interaction plausibility are ensured by iterative diffusion denoising and implicit policy optimization. (2) We propose to adjust the centroid of the sampling distribution during denoising process in a GAN Inversion manner for higher interaction plausibility, where the motion representation is designed to be fully differentiable with respect to the human mesh and joints. (3) We design to generate new motion based on historical constrains via inpainting and blend the motion in the power space of the rotation matrix using time-variant coefficients to synthesize long-term motion for multiple subsequent tasks.

## 2 RELATED WORK

**Human Scene Interaction.** Generating realistic and plausible human-scene interactions has been widely explored in the artificial intelligence and computer graphics communities Savva et al. (2014; 2016); Zhang et al. (2020a;b); Zhao et al. (2022). PLACE Zhang et al. (2020a) modeled the proximity based on the distance between the human body and the 3D scene to synthesize reasonable interactions. An optimization step was taken to adjust pose for plausible interactions under geometric constraints. PSI Zhang et al. (2020b) generated human bodies in 3D scenes conditioned on scene semantics and a depth map. Wang *et al.* Wang et al. (2021) generated a human body with pre-defined translation and orientation based on the scene point cloud. POSA Hassan et al. (2021b) designed a contact feature map for the human body, indicating the contact and semantic information for each vertex in the human mesh. COINS Zhao et al. (2022) utilized a Transformer-based generative network to encode the human body and 3D objects into a shared feature space and synthesizes diverse compositional interactions. Narrator Xuan et al. (2023) exploited the relationship between the 3D scene and the textual description based on a scene graph for interaction generation.

Inspired by the optimization stage in static human-scene interaction, we design interaction-based reward functions as an implicit policy for scene-aware motion synthesis.

**Motion Synthesis.** Motion synthesis is a long-standing problem that has been studied for a significant period Clavet et al. (2016); Holden et al. (2017); Starke et al. (2019). This topic has been researched conditioned on various signals, including motion prefixes Mao et al. (2019), actions Guo et al. (2020); Petrovich et al. (2021); Xu et al. (2023), music Gong et al. (2023); Tseng et al. (2022), and text Petrovich et al. (2022); Guo et al. (2022); Tevet et al. (2023). Action2Motion Guo et al. (2020) employed a recurrent conditional VAE for motion creation, where historical data was utilized to predict the subsequent pose. ACTOR Petrovich et al. (2021) encoded the entire motion sequence into a latent feature space, significantly reducing the accumulative error in recurrent methods. TEMOS Petrovich et al. (2022) utilized a VAE to learn a shared latent space for motion and textual description. The motion distribution and text distribution were well aligned by minimizing the KL divergence. T2M Guo et al. (2022) further trained a text-to-length estimator, enabling the network to automatically predict the length of the generated motion. MDM Tevet et al. (2023) and MotionDiffuse Zhang et al. (2024) are the pioneers to leverage diffusion model for human motion synthesis. Subsequent works Chen et al. (2023); Xie et al. (2023); Zhang et al. (2023b); Dai et al. (2024); Zhang et al. (2023c); Karunratanakul et al. (2023); Xie et al. (2024) further improved the controllability and quality of the generated results through database retrieval, spatial control, fine-grained captioning.

Thanks to the advancements in motion synthesis, we follow the MDM Tevet et al. (2023) and extend it to scene-aware motion synthesis, using interaction-based reward functions as an implicit policy.

**Scene-Aware Motion Synthesis.** Synthesizing realistic human motion in various scenes has garnered much attention in recent years Zhang et al. (2022); Hassan et al. (2023); Mir et al. (2024). Wang *et al.* Wang et al. (2021) utilized the PointNet Qi et al. (2017) to extract scene feature and optimized the entire motion based on the scene after generation. Wang *et al.* Wang et al. (2022a) brought more diversity into scene-aware motion synthesis by introducing three levels of diversity. SAMP Hassan et al. (2021a) utilized a mixture of expert networks to first predict the action state and then generate the motion. GAMMA Zhang & Tang (2022) modeled human pose using body markers and learned a latent space for plausible motion, where a policy network was later trained later to give appropriate motion under specific conditions. DIMOS Zhao et al. (2023) further introduced human-scene interaction and used PPO Schulman et al. (2017) to learn a policy network over a latent motion space. PAAK Mullen et al. (2023) placed human motion in scenes according to keyframe interactions. SceneDiffuser Huang et al. (2023) proposed a diffusion-based framework where a scene-conditioned diffuser is accompanied by a learning-based optimizer and planner to achieve the goal. LAMA Lee & Joo (2023) introduced a test-time optimization stage for controller network via reinforcement learning to predict the action cues for motion matching Clavet et al. (2016) and motion modification. AMDM Wang et al. (2024) designed a two-stage framework with a scene affordance map as an intermediate representation for final human motion synthesis.

Compared with these methods, we propose to disentangle scene-aware motion synthesis into motion prior learning via diffusion model and implicit policy learning via interaction-based reward functions, and integrate them in a unified framework, termed Diffusion Implicit Policy.

## 3 METHOD

### 3.1 PRELIMINARY

**Motion Representation.** For human motion, we take the SMPL-X model Pavlakos et al. (2019) to represent the pose at each frame. Here, we mainly consider the global orientation represented in axis-angle $\theta_{global} \in \mathbb{R}^3$, joint rotation in axis-angle $\theta_{j=1:21} \in \mathbb{R}^{63}$ and the translation $\tau \in \mathbb{R}^3$. Accordingly, for each frame $s$, the human pose can be defined as $P_s = \{\theta_{s,global}, \theta_{s,j=1:21}, \tau_s\} \in \mathbb{R}^{69}$, and the synthesized motion consisting of consequent poses can be annotated as $\hat{P} = \{\hat{P}_s\}_{s=1:S}$. The body shape $\beta \in \mathbb{R}^{10}$ and hand pose $\theta_h \in \mathbb{R}^{24}$ are always keep the same as initial human body for simplicity. The first $K$ joints $J = J_{1:K} \in \mathbb{R}^{K \times 3}$ and body mesh with $V$ vertices $M(\tau, \theta_{global}, \beta, \theta_j, \theta_h) \in \mathbb{R}^{V \times 3}$ are taken as auxiliary representation for human pose.

**Motion Diffusion Model.** For the diffusion model, we take motion with $S$ frames $x_0 = \{P_s\}_{s=1:S}$ as sample for training. The diffusion process will gradually add noise to the original motion $x_0$

$$q(x_t|x_{t-1}) = \mathcal{N}(\sqrt{\alpha_t}x_{t-1}, (1 - \alpha_t)I), \tag{1}$$

where $\mathcal{N}$ is a normal distribution and $\alpha_{t=1:T}$ are a series of hyper-parameters. The distribution of final noised motion $x_T$ will approximate to $\mathcal{N}(0, I)$. Following MDM Tevet et al. (2023), we choose to train a diffusion model to directly predict the original motion

$$\hat{x}_0^\phi = \phi(x_t, t, a), \tag{2}$$

where $t$ is the time step and $a$ is the action label for easier control of human motion behavior. As for motion synthesis, the denoising procedure can be formulated as:

$$P(x_{t-1}|\hat{x}_0^\phi, x_t) = \mathcal{N}(x_{t-1}; \mu_t(\hat{x}_0^\phi, x_t), \tilde{\beta}_t\mathbf{I}), \tag{3}$$

where $\tilde{\beta}_t = \frac{1 - \bar{\alpha}_{t-1}}{1 - \bar{\alpha}_t}\beta_t$, $\beta_t = 1 - \alpha_t$, $\bar{\alpha}_t = \prod_{i=1}^t \alpha_i$, and $\mu_t(\hat{x}_0^\phi, x_t) = \frac{\sqrt{\bar{\alpha}_{t-1}}\beta_t}{1 - \bar{\alpha}_t}\hat{x}_0^\phi + \frac{\sqrt{\bar{\alpha}_t}(1 - \bar{\alpha}_{t-1})}{1 - \bar{\alpha}_t}x_t$. Thanks to this formulation, we can easily adjust the distribution of denoised motion $P(x_{t-1})$ to pursue higher interaction plausibility in the final synthesized motion at each denoising step.

### 3.2 OVERVIEW

In this paper, we attempt to synthesize human motion in 3D scenes given a sequence of interaction sub-tasks (Fig. 2 (a)).

We can first decompose the whole command into a list of sub-task (interaction behavior and object pair) to be finished via current LLMs Achiam et al. (2023); Touvron et al. (2023). For each sub-task, the human may go to some place or interact with the objects in the scene (*e.g.* sitting on the chair).

Given a sub-task, we will first locate the goal position as COINS Zhao et al. (2022). Then, we will judge current action, and fetch feasible human-scene interaction reward functions accordingly.

We train a diffusion model conditioned on motion action and keyframe joints to synthesize natural motion with easy control, where body meshes and joints are fully differentiable for Diffusion Implicit Policy (Sec. 3.3). Later, we model the interaction-based reward functions (Sec. 3.4) and take them to optimize the sampling distribution of denoised motion at each denoising step (Sec. 3.5). The motion prior from diffusion model and implicit policy from reward function are integrated together to synthesize scene-aware motion with desired interactions (as shown in Fig. 2 (b)).

Given historical motion, the synthesized future motion should be in consistent with it. Thus, we design to derive the long-term motion via a time-variant blending (Sec. 3.6) where translation are interpolated linearly and rotation are blended in the matrix power space for motion transition. By now, we can synthesize long-term motion in 3D scenes when all sub-tasks are finished.

### 3.3 CONDITIONAL DIFFUSION MODEL

To ensure the body meshes and joints of synthesized motion are fully differentiable for Diffusion Implicit Policy, we directly train diffusion model $\phi$ to predict the original motion $\hat{x}_0^\phi$ for each noised $x_t$ following MDM Tevet et al. (2023) as shown in Eq. 2. We also take $x_0 \in \mathbb{R}^{S \times 69}$ consisting of human translation, orientation, and joint rotations in $S$ frames as the representation of motion.

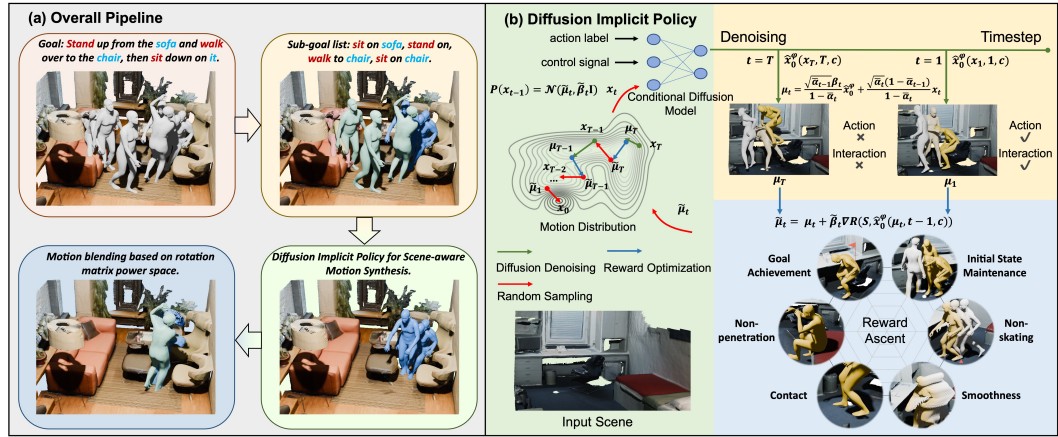

Figure 2: (a) indicates the overall pipeline of scene-aware motion synthesis. Any feasible command will be first decomposed into sub-task with action-object pairs. Then, we will synthesize the future motion according to history motion and current sub-task. Last, the synthesized motions will be fused into the history motion to obtain the final long-term motion. (b) presents the framework of Diffusion Implicit Policy (DIP). In each iteration of the DIP, the diffusion model will denoise the motion and enable the synthesized motion to appear more **natural**, and implicit policy optimization from reward will endow the motion with **plausible** interaction. The random sampling step can help the framework synthesize motion with **diverse** styles.

For any original motion $x_0$, we take the local coordinate of motion to reduce representation redundancy. Specifically, we translate and rotate the whole motion according to the human pose in the first frame. In the transformed motion, the human body in the first frame should be oriented towards the y-axis, with the top of the head facing the z-axis, and the pelvis positioned at the origin. Based on this setting, the motion in scenes can be derived via simple translation and horizontal rotation.

To assist the Diffusion Implicit Policy in maintaining the initial state and achieving goals, we also take a ControlNet branch Zhang et al. (2023a) to provide the hint of keyframe human joints (available from motion for training) that need to be controlled (Fig. 3 (b)). We choose to take the joint positions rather than joint rotation as the external hint for easier understanding of the space information in the diffusion model. Here, we mainly use the principle 22 joints from the SMPL-X human body skeleton as the skeleton joint hint. Concretely, we will first calculate the joints' positions according to the original human motion, and then randomly select one from these joints in a few frames as the hint and others are padded with zeros, thus the input of ControlNet branch take the form of $\mathcal{J} = \{J_{s,k}\}_{s=1:S,k=1:K}$ where non-zero values provide the controlled joint position hints.

For training the ControlNet branch, all its parameters are randomly initialized, and the link layers are initialized to zero to maintain the motion synthesis capability of the main branch. Meanwhile, all parameters in original motion diffusion model $\phi$ are frozen during training. The controlled diffusion model $\varphi$ is also supervised to predict the origin motion represented by $\{P_s\}_{s=1:S}$. After fine-tuning the ControlNet branch, the controlled motion diffusion model can be formulated as

$$\hat{x}_0^\varphi = \varphi(x_t, t, a, \mathcal{J}). \tag{4}$$

For simplicity, the action $a$ and joint hint $\mathcal{J}$ are termed as the condition $c$. Later, the controlled diffusion model $\varphi$ are taken as motion prior to optimize the noised motion for higher motion naturalness.

To better control the poses at specific frames, especially for the historical motion constrains, we maintain the fixed poses in an inpainting manner when $t > T_{inpaint}$. The predicted original motion $\hat{x}_0^\varphi$ will be updated via inpainting that can be formulated as $\hat{x}_0^\varphi = m \cdot x_m + (1 - m) \cdot \hat{x}_0^\varphi$, where $x_m$ indicates the poses that should be maintained and $m \in \mathbb{R}^{S \times 69}$ represents the inpainting mask.

Thus far, the diffusion model can provide motion prior with explicit joint control for later DIP.

### 3.4 REWARD DESIGN

Based on the diffusion model aforementioned, we further take the following reward functions as an implicit policy for scene-aware motion synthesis to pursue plausible human-scene interaction.

**Initial State Maintenance Reward.** To maintain continuity with historical motion $\tilde{P}_{1:\tilde{S}}$ when inferring future motion $\hat{P}_{1:S}$, the poses in the first $H$ frames of $\hat{P}_{1:S}$ should be consistent with $\tilde{P}_{\tilde{S}-H+1:\tilde{S}}$. Here, we choose to constrain the body joints to ensure the pose maintenance. Such reward can be formulated as:

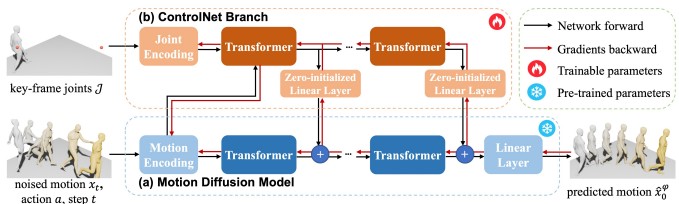

Figure 3: Illustration of conditional diffusion model. A diffusion model is first pre-trained conditioned on action, and then a ControlNet branch is introduced to provide keyframe joints' hint.

$$\mathcal{R}_{init} = \sum_{i=1}^{H} -|\hat{J}_i - \tilde{J}_{\tilde{S}-H+i}|. \tag{5}$$

**Goal Achievement Reward.** We also encourage the synthesized motion to achieve the goal, thus we take the goal position or pose joints in the form of $J_{goal}$ as the guidance. Here, we decide the frame $g$ that need to be controlled according to the distance and sampled speed (which is correlated with the action state). The goal achievement reward for the synthesized motion is defined as:

$$\mathcal{R}_{goal} = -|\hat{J}_g - J_{goal}|. \tag{6}$$

**Non-Skating Reward.** To avoid body skating during motion synthesis, we need to make sure that the velocity of contact body part to be close to $0$. Here, we decide to fetch all possible contact parts (*e.g.* feet during walking, gluteus and back during lying). Thus, the reward for non-skating can be formulated as follows:

$$\mathcal{R}_{skt} = \sum_{s=1}^{S-1} -ReLU(\min_c(||M_{c,s+1} - M_{c,s}||_2) \times \nu - \epsilon_{vel}), \tag{7}$$

where $M_{c,s}$ indicates the contacted parts of mesh vertices at frame $s$, $\nu$ is the Frame Per Second (FPS), and $\epsilon_{vel}$ is the tolerance for minor skating.

**Non-Penetration Reward.** For the human-scene interaction in the synthesized motion, penetration should be avoided. We utilize the scene Signed Distance Function (SDF) as the guidance, and such reward function will encourage the generated human body move away from the interior space of the scene mesh. The non-penetration reward take the form of

$$\mathcal{R}_{pene} = \sum_{s=1}^{S} \sum_{m} -ReLU(-f_{SDF}(M_{m,s}) - \epsilon_{pene}). \tag{8}$$

Here, $\epsilon_{pene}$ is the tolerance for slight penetration and we take SSM2 marker $M_m$ indicating 67 body surface vertices to simplify the mesh like previous works Zhang et al. (2021); Zhao et al. (2023).

**Contact Reward.** The synthesized motion should keep in contact with the scene. For locomotion, at least one foot vertices should be in touch with the floor, thus the contact reward is define as:

$$\mathcal{R}_{cont} = \sum_{s=1}^{S} -ReLU(|\min_f(M_{f,s}[z]) - h_{floor}| - \epsilon_{cont}), \tag{9}$$

where $M_f$ represents the foot vertices, $\epsilon_{cont}$ is the tolerance for contact. As for other actions where other parts of the body should be in contact with the scene, we also utilize the scene SDF for guidance where reward can be defined as:

$$\mathcal{R}_{cont} = \sum_{s=1}^{S} -ReLU(|\min_m(f_{SDF}(M_{m,s}))| - \epsilon_{cont}). \tag{10}$$

**Smoothness Reward.** During motion synthesis, we should also ensure the motion smoothness, and such reward is necessary when interaction-based implicit policy is taken into the diffusion model. Thus, we introduce a reward for limited acceleration in case of abrupt pose changes:

$$\mathcal{R}_{acc} = \sum_{s=2}^{S-1} (||M_{m,s+1} + M_{m,s-1} - 2M_{m,s}||_2 \times \nu^2 - \epsilon_{acc}). \tag{11}$$

Here, $\epsilon_{acc}$ is the maximum tolerant acceleration.

The total reward function for implicit policy take the form of

$$\mathcal{R}_{ip} = \lambda_{init}\mathcal{R}_{init} + \lambda_{goal}\mathcal{R}_{goal} + \lambda_{skt}\mathcal{R}_{skt} + \lambda_{pene}\mathcal{R}_{pene} + \lambda_{cont}\mathcal{R}_{cont} + \lambda_{acc}\mathcal{R}_{acc} \tag{12}$$

where $\lambda_{(\cdot)}$ are a series of hyper-parameters.

### 3.5 Diffusion Implicit Policy

In the denoising process, $x_{t-1}$ is sampled according to noised motion $x_t$, step $t$ and condition $c$. In this paper, we utilize a diffusion model with ControlNet branch $\varphi$ to predict the origin motion $\hat{x}_0^\varphi$. Thus, similar to Eq. 3, we can sample the $x_{t-1}$ according $x_t$ and $\hat{x}_0^\varphi$

$$P(x_{t-1}|\hat{x}_0^\varphi, x_t) = \mathcal{N}(x_{t-1}; \mu_t(\hat{x}_0^\varphi, x_t), \tilde{\beta}_t \mathbf{I}). \tag{13}$$

Such denoising process can also be considered as an optimization problem based on a motion naturalness reward function $R_{nat}$ which can be defined implicitly by its gradient $\nabla R_{nat}(x_t) = \mu_t - x_t$. Meanwhile, the denoising process is also accompanied by a stochastic disturbance item following $\mathcal{N}(0, \tilde{\beta}_t \mathbf{I})$.

Thus, the stochastic item can be well utilized to search for motion with higher interaction plausibility. We design the interaction-based implicit policy to partly play the role of such disturbance, and the whole scene-aware motion synthesis can be treated as a joint optimization problem which both maximize the motion naturalness and interaction plausibility for Diffusion Implicit Policy

$$\hat{x}_0 = arg \max_x \mathcal{R}_{dip}(x), \quad \mathcal{R}_{dip}(x) = \mathcal{R}_{nat}(x) + \mathcal{R}_{ip}(\hat{x}_0^\varphi(x)). \tag{14}$$

In order to synthesize human motion that can maximize the total reward, we integrate the implicit policy optimization into each denoising step where motion naturalness and interaction plausibility can be enhanced iteratively.

We can observe that in Eq. 13, $x_{t-1}$ is sampled from $\mathcal{N}(\mu_t, \tilde{\beta}_t \mathbf{I})$, thus we can adjust $\mu_t$ (*i.e.* the mean value of $x_{t-1}$) based on implicit policy and more suitable $x_{t-1}$ can be sampled accordingly. It is noteworthy, we need the final synthesized human motion $x_0$ to be consistent with the scene and achieve high reward. Thus, we propose to optimize $\mu_t$ through $\hat{x}_0^\varphi(\mu_t, t-1, c)$ rather than $\mu_t$ itself. Here, we take $t-1$ as denoising step because $\mu_t$ is the mean value of the denoised $x_{t-1}$.

Thanks to the motion representation taken in Sec. 3.3, the reward functions are fully differentiable. Meanwhile, we find that optimizing $\hat{x}_0^\varphi(\mu_t, t-1, c)$ shows better performance than directly modifying $\mu_t$ itself as previous work Xie et al. (2024). That is because direct optimization over $\mu_t$ does not ensure motion continuity. On the other side, optimizing $\mu_t$ via $\hat{x}_0^\varphi(\mu_t, t-1, c)$ can help search a better distribution with higher interaction reward for the final synthesized motion $x_0$, and $\mu_t$ can be be adjusted as a whole in a GAN Inversion manner (adjusting latent code $z$ through the Generator G(z) to keep reality) Bau et al. (2019a;b). Thus, we optimize $\mu_t$ according the following formulation

$$\tilde{\mu}_t = \mu_t + \tilde{\beta}_t \cdot \nabla \mathcal{R}_{ip}(\mathcal{S}, \hat{x}_0^\varphi(\mu_t, t-1, c)), \tag{15}$$

where $\mathcal{S}$ indicate the 3D scene information, including scene semantics, SDF, and floor height. Further, $\tilde{\mu}_t$ is taken as the mean of distribution to sample $x_{t-1}$.

### 3.6 Multi-Task Motion Synthesis

As for the command for multi-task motion synthesis, such as "The person first sits on the bed, then goes to the corner of the room, and finally sits on the chair.", we need to infer future motion "Sit on the chair" while maintaining continuity with previously synthesized motions "The person first sits on the bed and then goes to the corner of the room".

For any previous synthesized motion $\tilde{P}_{1:\tilde{S}}$, we will select the latest $H = min(\tilde{S}, H_{max})$ frames as external historical constrains for future motion synthesis. Explicitly, we extract those pelvis joints $\{\tilde{J}_{s,pelvis}\}_{s=\tilde{S}-H+1:\tilde{S}}$ as trajectory hints for conditional motion diffusion. In addition, we take the body skeleton joints from the historical frames to form $\mathcal{J} = \{\tilde{J}_s\}_{s=\tilde{S}-H+1:\tilde{S}}$, and use them for pose constraints in the implicit policy optimization.

After a new round of motion synthesis, we obtain the generated motion $\{\hat{P}_s\}_{s=1:S}$. For more natural motion transition in the overlapping $H$ frames, we take a time-variant motion blending. Different from direct linear interpolation in the pose representation like priorMDM Shafir et al. (2024). We only utilize linear interpolation for the human translation $\check{\tau}_{1:H}$. As for human orientation and joints' rotations, we take the axis-angle representation $\bar{\theta}_{rot,s} = \tilde{\theta}_{\tilde{S}-H+s}$ and $\hat{\theta}_{rot,s}$ and convert them to the

rotation matrix $\bar{M}_{rot,s}$ and $\hat{M}_{rot,s}$. The blending occur in the power space of rotation matrix which can be formulated as:

$$\check{M}_{rot,s} = (\hat{M}_{rot,s}\bar{M}_{rot,s}^{-1})^{\gamma}\bar{M}_{rot,s} \tag{16}$$

where $\gamma = (1:H)/(H+1)$, and $\check{\theta}_{1:H}$ is converted from $\check{M}_{rot,1:H}$. Thus, the blended pose take the form of $\check{P}_{1:H} = \{\check{\theta}_{s,global}, \check{\theta}_{s,j}, \check{\tau}_s\}_{s=1:H}$. The newly updated long-term motion is derived as

$$\tilde{P}_{1:\tilde{S}+S-H} = \{\tilde{P}_{1:\tilde{S}-H}, \check{P}_{1:H}, \hat{P}_{H+1:S}\}. \tag{17}$$

The whole motion can be synthesized in an iterative manner until all the tasks are finished.

## 4 EXPERIMENTS

### 4.1 DATASETS

**Motion Datasets.** Here, we use captured motion data from AMASS Mahmood et al. (2019), which include action/description labels, to train our controlled motion diffusion model. Babel Punnakkal et al. (2021) provided action labels and the start/end frames for several subsets of AMASS. HumanML3D Guo et al. (2022) provided additional sentence annotations and start/end frames for more motion data in AMASS. We match the keywords and categorize them into three states where details are presented in the Appendix. All motions are downsampled to 40 FPS and split into 160-frame motion clips with a 20-frame stride. Motion clips are all transformed according to the human pose in the first frame, where the transformed initial pose is centered (with the pelvis located at the origin) and oriented towards the y-axis with z-axis up. All the motion clips, along with the action labels, are used to train the motion diffusion model. Additionally, skeletons are extracted to provide joint position hints when training the ControlNet branch.

**Scene Datasets.** We evaluate the performance of the proposed Diffusion Implicit Policy framework in both synthesized scenes and real scanned scenes. Following DIMOS Zhao et al. (2023), we use randomly generated scenes consisting of furniture from ShapeNet Chang et al. (2015) to validate the performance on atomic locomotion and human-scene interaction. As for real scanned scenes from PROX Hassan et al. (2019) and Replica Straub et al. (2019),

Table 1: Evaluation of motion synthesis on locomotion task. The up/down arrows ($\uparrow$/$\downarrow$) indicate higher/lower is better. Metrics with best performance are annotated in boldface.

|  | time $\downarrow$ | avg. dist $\downarrow$ | contact $\uparrow$ | loco pene $\uparrow$ |
|---|---|---|---|---|
| SAMP | 5.97 | 0.14 | 0.84 | 0.94 |
| GAMMA | 3.87 | **0.03** | 0.94 | 0.94 |
| DIMOS | 6.43 | 0.04 | **0.99** | **0.95** |
| Ours | **3.35** | **0.03** | 0.91 | **0.95** |

we take them to evaluate the performance of the pipeline in synthesizing long-term motions within scenes that involve multiple tasks. All the experiments are conducted using the same controlled motion diffusion model and pipeline, thus indicating the generalization ability of the proposed method.

### 4.2 SCENE NAVIGATION

We take the generated scenes from DIMOS Zhao et al. (2023) for testing, where scenes are cluttered with furniture from ShapeNet Chang et al. (2015). In this experimental setting, the human need to walk from the starting point to the target point and avoid collisions with the furniture in scenes.

**Metrics.** We also evaluate the performance of synthesized motion for locomotion from four aspects as DIMOS Zhao et al. (2023), where (1) finish time measured in seconds, (2) average horizontal distance from final human body to target point measured in meters, (3) foot joint contact score

$$s_{contact} = e^{-(|minj_z|-0.05)_+} \cdot e^{-(min||j_{vel}||_2 - 0.075)_+}, \tag{18}$$

and (4) locomotion penetration score indicating the percentage of body vertices within the walkable area are taken as our metrics.

**Results.** We compare our method with SAMP Hassan et al. (2021a), GAMMA Zhang & Tang (2022), and DIMOS Zhao et al. (2023) for locomotion in 3D scenes and report the results in Tab. 1. The results indicate our method can achieve the shortest finish time (3.35s), closest distance (0.03m), and lowest penetration (0.95). It's noteworthy, the locomotion speed of the proposed method is much similar to that of real human than other methods. As can be seen, the performance on the contact score is inferior. We believe that is because our method focuses more on foot vertex contact, whereas the contact score calculation is based on foot joints.

We visualize a few examples of DIMOS and the proposed for locomotion task in Fig. 4. As shown, the synthesized motion for locomotion demonstrates lower scene penetration, less skating and higher motion diversity. That is attributes to the implicit policy and the stochastic sampling procedure during denoising.

### 4.3 SCENE OBJECT INTERACTION

We take scenes with furniture from ShapeNet Chang et al. (2015) to evaluate the performance of the proposed method on scene object interaction, and 10 objects (3 armchairs, 3 straight chairs, 3 sofas and 1 L-sofa) are chosen for atomic interaction as previous work Zhao et al. (2023). For each scene, the human is initialized to stand in front of the interaction object and guided to interact with it and finally return to original position.

**Metrics.** We take 4 metrics to evaluate the performance of interaction, including (1) the time to finish the task measured in seconds, (2) the foot contact score mentioned in Eq. 18, (3) mean human mesh vertex penetration $\sum_{v \in M} |(f_{SDF}(v))_-|$ over time, and (4) maximum penetration over time.

**Results.** We compare the proposed method with prevailing methods Hassan et al. (2021a); Zhao et al. (2023) on human-scene interaction. The interaction tasks for sitting and lying are evaluate separately, and the results are reported in Tab. 2.

Fig. 5 visualizes the synthesized human-scene interaction given by DIMOS and the proposed method. The visual results indicate the proposed present more plausible interaction with higher human-scene contact and lower human-scene collision.

Table 2: Evaluation of motion synthesis on interaction tasks. The up/down arrows (↑/↓) indicate higher/lower is better. The best results are shown in boldface.

|  | time ↓ | contact ↑ | pene. mean ↓ | pene. max ↓ |
|---|---|---|---|---|
| SAMP sit | 8.63 | 0.89 | 11.91 | 45.22 |
| DIMOS sit | 4.09 | **0.97** | 1.91 | 10.61 |
| Ours sit | **3.71** | 0.89 | **1.86** | **7.13** |
| SAMP lie | 12.55 | 0.73 | 44.77 | 238.81 |
| DIMOS lie | 4.20 | **0.78** | 9.90 | 44.61 |
| Ours lie | **3.55** | 0.68 | **9.80** | **30.8** |

Figure 4: Visual results given by DIMOS and our method for locomotion task. The dashed circles indicate lower penetration, less skating and higher diversity in the synthesized motion.

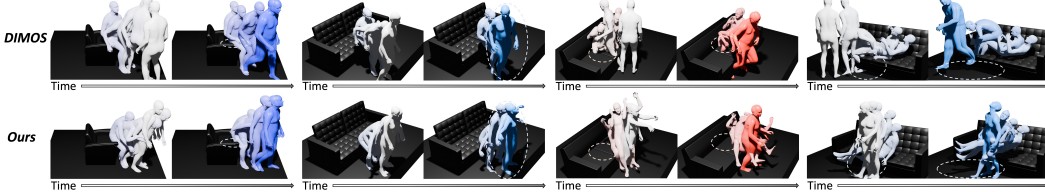

Figure 5: Visual results of synthesized motions given by DIMOS and our method for sitting (left) and lying (right) task. The dashed circles indicates obvious advantages over DIMOS in less collision (col. 1,3), higher motion diversity (col. 2) and better foot contact (col. 4).

### 4.4 LONG-TERM MOTION SYNTHESIS

For long-term motion synthesis in 3D scenes where multiple tasks are completed consecutively, objects with feasible interaction in scenes are randomly selected. We utilize COINS Zhao et al. (2022) to sample the static interactions with these objects as the goals. All compared methods use the same initial state and task goals to synthesize long-term motions, ensuring a fair comparison.

**Metrics.** To better evaluate the motion naturalness, diversity, interaction plausibility and overall performance, we conduct a user study where the synthesized motions given by different methods are directly judged by humans. We totally generate 20 motion for each method, 10 for scenes from PROX, and 10 for scenes from Replica. We present motions synthesized by different methods in the same scene to the participants simultaneously, and ask the users to rate the results accordingly.

**Results.** Finally, $1,200$ ratings are collected for each method (60 results per sample from 4 aspects) and we report the scores in Tab. 3. It can be seen, the proposed method achieve the best performance

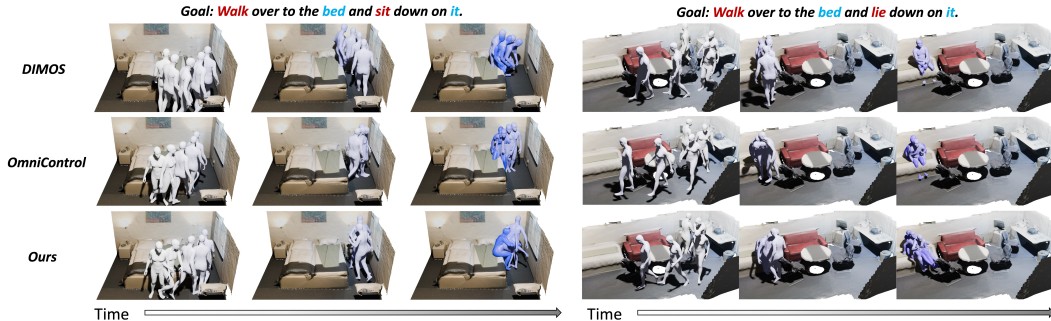

Figure 6: Visual comparison of different methods on motion synthesis in 3D scenes from PROX.

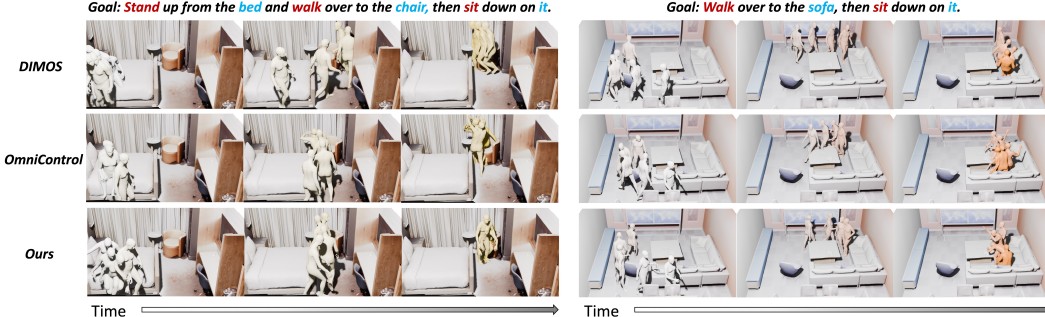

Figure 7: Visual results of synthesized motions given by compared methods in Replica scenes.

in motion diversity, interaction plausibility and overall performance even with no need of paired motion-scene data for training. For motion naturalness, our proposed method is on par with DIMOS, as no specific designs are taken to further improve motion naturalness.

We also show the results of scene-aware motion synthesis in scenes from PROX and Replica dataset respectively in Fig. 6 and Fig. 7 (please refer to the Appendix for additional visual results and the Supplementary Material for a video). It can be seen that the proposed Diffusion Implicit Policy performs well in

Table 3: Comparison between competitive methods based on user study. Users are asked to give scores (ranging from 1 to 5, ↑) according to motion naturalness, diversity, interaction plausibility, and overall performance. Results on PROX/Replica are reported on the left/right respectively.

| Scores / Methods | Naturalness ↑ | Diversity ↑ | Plausibility ↑ | Overall ↑ |
|---|---|---|---|---|
| DIMOS | 2.72/**3.09** | 3.00/3.21 | 2.55/2.85 | 2.86/3.11 |
| OmniControl | 2.66/2.67 | 2.83/3.07 | 2.26/2.5 | 2.61/2.69 |
| Ours | **2.81**/3.03 | **3.34/3.36** | **3.15/3.17** | **3.17/3.26** |

terms of motion naturalness, interaction plausibility and motion diversity thanks to the integration of motion denoising, implicit policy optimization, and random sampling within a unified framework.

## 5 CONCLUSION

**Conclusion.** In this paper, we propose a unified framework, termed Diffusion Implicit Policy, for long-term motion synthesis in 3D scenes. In this framework, interaction is disentangled from motion learning during training, and motion prior from diffusion model and implicit policy from interaction-based reward are integrated together to iteratively optimize the motion from random noise, pursuing motion naturalness, diversity and interaction plausibility simultaneously. We utilize joint hints and inpainting to ensure that keyframe poses remain consistent with the historical motion. We adjust the sample distribution centroid in a GAN Inversion manner achieve better interaction plausibility while maintaining motion continuity. We introduce motion blending in the power space of the rotation matrix and the linear space of translation with time-variant coefficients to ensure smooth transitions between multiple tasks for long-term motion synthesis. Comprehensive experiments on generated scenes with ShapeNet furniture, and scenes from PROX and Replica indicate the effectiveness and generalization capability. This paper provides a promising solution for synthesizing scene-aware motion without the need for paired motion-scene data during training. It also encourages future works to learn from unpaired motion and scene data.

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

# A APPENDIX

## A.1 IMPLEMENTATION DETAILS

**Motion Synthesis Details.** The conditional motion diffusion model is designed to synthesize motion lasting $S = 160$ frames within $T = 10^3$ steps. The first $K = 22$ joints from SMPL-X model are selected to represent the body skeleton. Last $H$ (at most $H_{max} = 10$) frames are taken as historical hints for motion blending. The keywords for action state categorization are shown in Tab. 4.

**Reward Function Details.** Commonly, we set $\epsilon_{vel} = 0.5$, $\epsilon_{pene} = 0.03$, $\epsilon_{cont} = 0.01$, and $\epsilon_{acc} = 50$. For reward functions, $\lambda_{init}$ and $\lambda_{goal}$ are set to 1, and $\lambda_{cont}$ is set to $10^{-1}$. For locomotion, $\lambda_{skt}$ is set to $10^{-3}$. For sitting, $\lambda_{skt}$, $\lambda_{pene}$, and $\lambda_{acc}$ are set to $3 \times 10^{-4}$, $10^{-1}$, and $10^{-3}$. For lying, $\lambda_{pene}$ and $\lambda_{acc}$ are set to $3 \times 10^{-2}$ and $10^{-3}$ respectively. All other coefficients are set to 0.

Table 4: Keywords used to match the action labels or sentence annotation in Babel and HumanML3D, and the matched motions are categorized into specific actions.

| Action | locomotion | sit | lie |
|---|---|---|---|
| Keywords | walk, turn, jog, run | sit | lie, lying |

## A.2 More Visual Results

We present additional visual results on PROX and Replica in Fig. 8 and Fig. 9. Thanks to integration of motion denoising, implicit policy optimization, and random sampling within the proposed Diffusion Implicit Policy framework, motion naturalness, interaction plausibility and motion diversity can be obtained simultaneously.

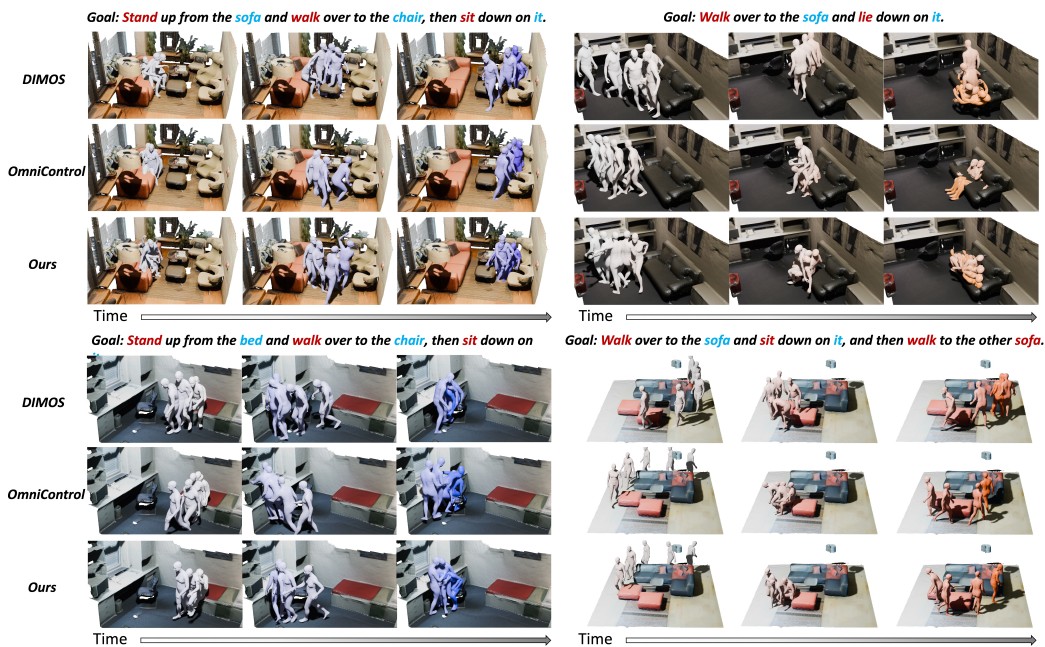

Figure 8: More visual comparisons in 3D scenes from PROX.

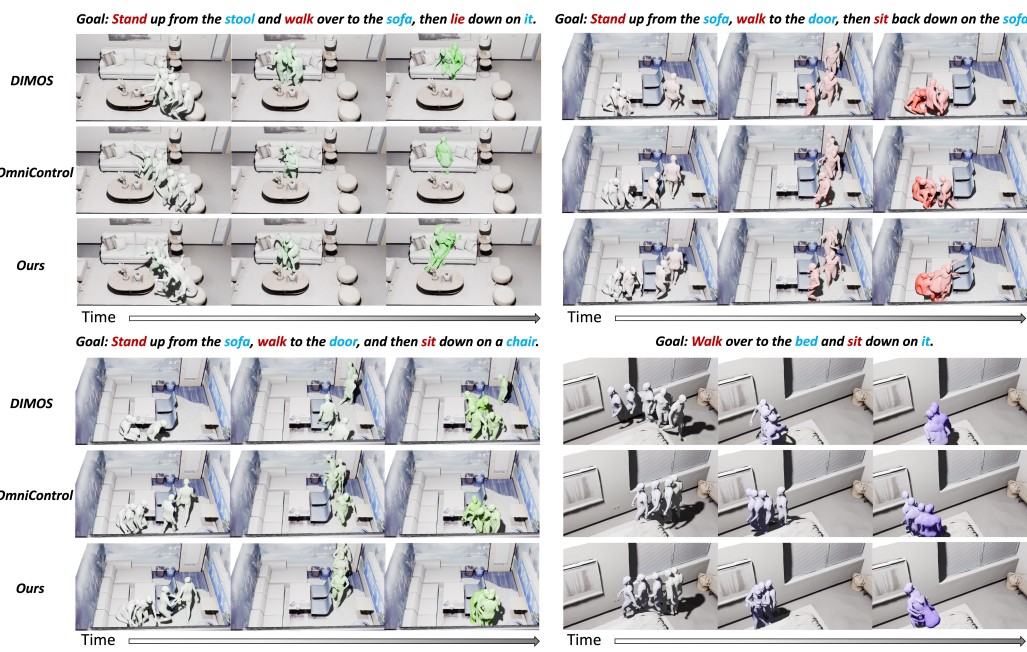

Figure 9: Visual results of additional synthesized motions in 3D scenes from Replica dataset.

Table 5: Average distance to desired goal position for synthesized interaction motions with/without motion inpainting.

|              | distance (sit ↓) | distance( lie ↓) |
| ------------ | ---------------- | ---------------- |
| w/o inpainting | 0.14           | 0.11             |
| w/ inpainting  | **0.08**       | **0.09**         |

### A.3  ABLATION STUDY

In this section, we conduct more experiments to validate the proposed framework and prove our claims.

#### A.3.1  DIRECT OPTIMIZATION V.S. GAN INVERSION

In the implicit policy optimization, we propose to optimize $\mu_t$ via $\hat{x}_0^\varphi(\mu_t, t-1, c)$ rather than directly optimize $\mu_t$ as OmniControl Xie et al. (2024). In this way, a better sample distribution centroid $\mu_t$ can be searched in GAN Inversion manner to satisfy the interaction in final synthesized motion $\hat{x}_0^\phi$. In addition, $\mu_t$ can be optimized as a whole instead of optimizing only one pose in one frame, thus can better keep motion continuity. Here, we compare the synthesized motions of these two strategies to prove the advantage of the proposed framework on final motion naturalness. Fig. 10 illustrates the comparisons in a scene from PROX dataset, and we mark the synthesized motions with discontinuity in red/green circles. Without motion searching through GAN Inversion, the reward will directly guide the motion distribution centroid $\mu_t$, and we can see the sparse constrains in reward function will definitely make synthesized motions have abrupt changes.

#### A.3.2  INPAINTING

In order to keep consistent with the historical motion and better achieve the task goal, we decide to maintain the poses in key frames that need to be controlled in an inpainting manner. We synthesize motion in scenes with ShapeNet furniture with/without the keyframe pose inpainting, and judge the performance according to the goal achievement. We report the average distance to the desired goal position in Tab. 5. As can be seen, the human can get closer to the destination when the motion are denoised with inpainting for human-scene interaction.

Figure 10: Comparison between two optimization strategy. For each pair, the left sub-figures show the results given by direct optimization, and the right sub-figures present the synthesized motions derived from optimization in GAN Inversion manner. Motions with obvious discontinuity are marked in red/green dashed circles.

### A.4  DISCUSSION

According to the experiments in this paper, we could see entangling motion diffusion model and interaction-based implicit policy makes full utilization of the stochastic procedure in motion denoising, and can outperform current explicit policy method even without paired motion-scene data for training. This also indicates the proposed Diffusion Implicit Policy can generalize to diverse scenes as no specific scenes are required for training.

Even though, there are still some limitations in current method. As the implicit policy is introduced into motion denoising with limited gradient scale, there is still occasional collision between human and scene. Meanwhile, the motion style cannot be controlled currently, where action may be replaced by command in future work for easier style control.

