# OpenReview forum: "Diffusion Implicit Policy for Unpaired Scene-aware Motion Synthesis"
_ICLR.cc/2025/Conference — ICLR 2025 Conference Withdrawn Submission_

### Official Review · Reviewer_GnpB · 2024-10-28

**Soundness:** 3
**Presentation:** 3
**Contribution:** 2
**Rating:** 5
**Confidence:** 3

**Summary:**

This paper proposes a unified framework for synthesizing scene-aware human motion under Diffusion Implicit Policy (DIP). The DIP divides itself with previous mehtods by disentagling human-scene interaction and motion synthesis during training to cope with limited motion-scene data.

**Strengths:**

1. This paper tackles inherent probelm of scene-aware motion synthesis of limited motion-scene paired data.
2. The proposed DIP achieves promising results in penetration and time metrics.

**Weaknesses:**

1. Overall writing of the paper is mostly not self-contained. This means that readers with limited background of diffusion-based motion synthesis may have difficult time comprehending the gist of the paper. Most importantly, it is difficult to understand why the method is free from motion-scene paired data.
2. There is no comparison on quality of generated motions with famous metrics such as FID or action recognition accuracy.
3. The proposed DIP fall short in contact metric as shown in Table 1 and Table 2.

**Questions:**

1. May I ask for further explanation regarding L188 - L189? Without any further explanation or at least references to explain the claim, it is difficult to comprehend that this formulation naturally pursue higher interaction plausibility in final synthetized motion.
2. What does it mean to “optimize the intermediate noised motion in a GAN Inversion manner? While this is referred multiple times within the manuscript, no specific explanation was done. It is unclear “How” GAN inversion was conducted not “Why” GAN inversion is superior than direct optimization.
3. How was the user study conducted? There should be more detailed explanation regarding how the user study was done and number of participants for the user study.

---

### Official Review · Reviewer_YTym · 2024-10-31

**Soundness:** 3
**Presentation:** 2
**Contribution:** 2
**Rating:** 5
**Confidence:** 4

**Summary:**

This paper presents a framework, namely Diffusion Implicit Policy, for scene-aware motion generation. To address the reliance on scene-motion paired dataset, DIP disentangle the interaction and motion generation during training. Specifically, the motion generation part is formulated as a conditional motion diffusion model and is trained on large-scale motion-only dataset to provide high quality motion priors. Human-scene interaction is formulated as reward functions, which act as optimization targets during diffusion sampling. Motion blending is also designed for smooth long-term motion generation. Experiments on synthesis scene and real scene show the effectiveness of the proposed method.

**Strengths:**

+ Compared with two baseline methods, the generated motion sequence is apparently more natural and physically plausible, with less penetration with the scene and less foot skating.
+ The idea of using diffusion model to improve motion quality is reasonable and might be the trend.

**Weaknesses:**

- The effectiveness of the ControlNet structure of the motion diffusion model is questionable as there is no ablation study. The position control should be able to enforce during diffusion sampling. Why bother training a control module?
- The interaction policy consists of well known constraint functions, making the contribution of this part weak. It is also debatable whether these constraint terms can be called reward, which is a taxonomy in the reinforcement learning.
- From the visual results, the generation motions seem to have unreasonable arm movements. Is this problem caused by inference-time optimization?
- Writing issues:
1. Almost all the citation format is wrong. Use \citep{} instead of \cite or \citet, (e.g., AMASS~\citep{amass})
2. Line 197: e.g. => e.g.,
3. Line 328: according => according to

**Questions:**

See the weaknesses section.

---

### Official Review · Reviewer_VKGW · 2024-11-04

**Soundness:** 2
**Presentation:** 2
**Contribution:** 2
**Rating:** 3
**Confidence:** 4

**Summary:**

This paper presents a method for generating SMPL-X human motions for scenes, conditioned on past poses and/or language instructions. The key idea is to set this up as a diffusion problem, guided with a ControlNet. The motion prediction objective is complemented with 6 handcrafted reward functions, which help to make the motion trajectory realistic. The proposed approach appears to outperform competitive baselines for the task.

**Strengths:**

Optimizing the main objective alongside 6 reward functions seems very tricky, and I am impressed that this worked.

**Weaknesses:**

This paper seems to have logical or grammatical or formatting errors throughout, relating to how citations belong in sentences. For example it often calls specific people "works", and names and semicolons are often inside sentences where they should not be part of the prose. E.g., "Some of them Wang et al. (2021; 2022a) utilized..." -- this doesn't make sense. Maybe the authors simply pasted their text into the ICLR template and then did not check for any errors. This makes the paper much more difficult to read than it should be.


The paper says that its main contribution is "We propose a brand-new framework,termed Diffusion Implicit Policy, for scene-aware motion synthesis." To me, the newness of the method, and the name of the method, are not very appealing contributions. I recommend that the authors state, as their primary contribution, something more useful about their work.


The authors state that "The diffusion process will gradually add noise to the original motion x0". This is surprising, because a very useful property of the diffusion formulation is that you can jump directly to a desired noise level, instead of stepping there gradually. Why do this? It seems like the method can be sped up a lot by using the standard method.



"given a sequence of interaction sub-tasks" -- not sure what this means.



"Given historical motion, the synthesized future motion should be in consistent with it" -- Do the authors mean inconsistent, or consistent?

The authors say "To ensure the body meshes and joints of synthesized motion are fully differentiable for Diffusion Implicit Policy, we directly train diffusion model ϕ to predict the original motion ˆxϕ0 for each noised xt following MDM Tevet et al. " -- it's not clear to me what the connection is between this detail and differentiability. Are the authors saying that if you predict the epsilon, it won't be differentiable?

The authors say that by normalizing the data by the zeroth pose, "motion in scenes can be derived via simple translation and horizontal rotation". This does not sound true. In general we seek much more detailed motion, and even the figures show the sort of motion I am imagining. Also, I am skeptical of the "horizontal rotation" -- rotation about any horizontal axis seems far too acrobatic for the task considered here.

In two places, it says that the controlnet-informed diffusion model is trained to predict "the origin motion". What is the origin motion? It is referred to with different symbols in the two locations, x_0^psi and {P_s}.

The "initial state maintenance" reward seems to ask for the body to be perfectly still. Why is this a good thing?

It seems like multiple modules are introduced twice: once in 3.3, and once in 3.5. (This is in addition to the initial introductions in 3.2.) Am I misunderstanding something here -- e.g., could it be that these are actually different modules (but with the names)?

In the multi-task motion synthesis, it is not clear to me how it is determined when a task is "finished".


How is the model optimized? From these "reward" functions it seems like reinforcement learning is involved, but I couldn't find anything concrete on this topic.







consequent -> subsequent

**Questions:**

Please interpret my list of "weaknesses" as mostly questions.

---

### Official Review · Reviewer_zxZ8 · 2024-11-04

**Soundness:** 3
**Presentation:** 2
**Contribution:** 2
**Rating:** 5
**Confidence:** 4

**Summary:**

This paper targets scene-aware human motion synthesis under the assumption that there are no paired motion-scene data. It disentangles human-scene interaction from motion synthesis during training and then leverages an interaction-based implicit policy that regulates motion diffusion during inference for motion synthesis. Specifically, the implicit policy optimizes the intermediate noisy motion in a GAN Inversion to maintain motion continuity and control keyframe poses through the Controlnet and motion inpainting. Motion blending is used to ensure smooth synthesis for long-term motion. The whole synthesis procedure can be thought of as an optimization problem to ensure natural motion while making the interaction compatible with the scene and task. Evaluations are performed on both simulated and real scenes. It shows promising results compared to the baselines in terms of motion naturalness and interaction plausibility.

**Strengths:**

There are several strengths of the proposed method. First, by leveraging the implicit diffusion policy, motion synthesis is converted into an optimization problem instead of supervised learning that requires paired data of motions and scenes, which can efficiently alleviate the difficulty induced by the lack of training data, which be able to ensure the naturalness of the motion as long as the rewards are designed effectively. Second, the introduced centroid-oriented sampling distribution in the denoising process under GAN inversion can effectively guarantee plausibility. Moreover, the generation depends on historical data and leverages inpainting to blend the motion in transitions, these guarantee the continuity of long-term motion synthesis.

**Weaknesses:**

There are also several weaknesses in the proposed. First, it does not model the interaction between the hands and the objects, which restricts the motion synthesis to scenarios where one only emphasizes correct contact of the body shape with coarse surfaces in the scene, which further limits the diversity of the synthesized motion. Second, the dependence on human motion diffusions on one hand provides human motion priors for motion synthesis, but on the other hand, can be a source of bias, so it is critical to establish the solidity of the proposed using multiple scene-unaware motion synthesis modules. More specifically, why was the motion diffusion model picked in the paper, how about other models? Thirdly, The employed reward functions are also observed in the literature, it is more or less like an inclusion of all of them, which may not be bad, but currently, it does not contribute too much to the novelty of the proposed pipeline. Furthermore, the whole system looks very heavy, e.g., too many components, and also the implicit policy optimization is used in each denoising step, which makes it inefficient, so how does the robustness look like, and how it is practically useful?

**Questions:**

The initial state maintenance reward is a bit weird, can you elaborate on the reasoning?
The motion blending does not look much different from priorMDM, anything new?

---

### Note · Authors · 2024-11-14

I have read and agree with the venue's withdrawal policy on behalf of myself and my co-authors.